# Effect of Hyperprolactinemia on Bone Metabolism: Focusing on Osteopenia/Osteoporosis

**DOI:** 10.3390/ijms25031474

**Published:** 2024-01-25

**Authors:** Soo Jin Yun, Hyunji Sang, So Young Park, Sang Ouk Chin

**Affiliations:** Department of Endocrinology and Metabolism, Kyung Hee University College of Medicine, Kyung Hee University Hospital, Seoul 02447, Republic of Korea; soojinyun@khu.ac.kr (S.J.Y.); roselorenz322@hotmail.com (H.S.); malcoy@hanmail.net (S.Y.P.)

**Keywords:** hyperprolactinemia, bone metabolism, osteoporosis, osteopenia

## Abstract

Prolactin is a hormone secreted from lactotroph cells in the anterior pituitary gland to induce lactation after birth. Hyperprolactinemia unrelated to lactation is a common cause of amenorrhea in women of a childbearing age, and a consequent decrease in the gonadotropin-releasing hormone (GnRH) by a high prolactin level can result in decreased bone mineral density. Osteoporosis is a common skeletal disorder characterized by decreased bone mineral density (BMD) and quality, which results in decreased bone strength. In patients with hyperprolactinemia, changes in BMD can be induced indirectly by the inhibition of the GnRH–gonadal axis due to increased prolactin levels or by the direct action of prolactin on osteoblasts and, possibly, osteoclast cells. This review highlights the recent work on bone remodeling and discusses our knowledge of how prolactin modulates these interactions, with a brief literature review on the relationship between prolactin and bone metabolism and suggestions for new possibilities.

## 1. Introduction

Hyperprolactinemia is typically defined as a fasting serum prolactin level > 20 ng/mL in men and 25 ng/mL in women [1]. The prevalence of hyperprolactinemia varies widely depending on the study, with the reported rates ranging from 0.4% in the general population to 9–17% in reproductive-age women [2,3]. Hyperprolactinemia can occur physiologically due to pregnancy, stress, or nipple stimulation [4,5]. Moreover, the pathological causes of hyperprolactinemia include prolactin-producing pituitary adenomas, hypothalamic/pituitary tumors, hypothalamic infiltrative diseases, and hypothyroidism [6]. In addition, patients with chronic kidney disease can experience elevated prolactin levels because of reduced prolactin clearance and changes in central prolactin regulation [7]. Furthermore, prolactin levels can be elevated by medications, such as antipsychotics including haloperidol or risperidone; antidepressants; antiemetics, such as metoclopramide or domperidone; H_2_ antihistamines; and cholinergic agonists [8,9,10,11] (Table 1).

Hyperprolactinemia is one of the most common causes of amenorrhea in women of a reproductive age [2]. Therefore, prolactin levels are measured as part of the diagnostic evaluation of women experiencing amenorrhea and in those experiencing oligomenorrhea, galactorrhea, or infertility, as well as of men with hypogonadism or erectile dysfunction. Together, elevated prolactin levels and amenorrhea are associated with the suppression of the gonadotropin-releasing hormone (GnRH), which leads to the reduced secretion of the luteinizing hormone (LH) and follicle-stimulating hormone (FSH) from the anterior pituitary gland. As a result, similar to other cases of secondary hypogonadism, serum gonadotropin levels normalize or decrease, leading to, among other consequences, a decrease in bone mineral density (BMD) (Figure 1). In men, symptoms of hyperprolactinemia can include a decreased libido, oligospermia, azoospermia, erectile dysfunction, infertility, gynecomastia, and, occasionally, galactorrhea. Reduced testosterone levels can also lead to decreased bone density [12].

Osteoporosis is a common skeletal disorder characterized by reduced bone strength due to decreased BMD and bone quality. In postmenopausal women, osteoporosis is defined as a condition where the T-score, comparing BMD to the average value in young adults, is ≤−2.5. In the United States, approximately 10.2 million adults aged 50 years and older have been diagnosed with osteoporosis [13]. Furthermore, owing to an aging population, the prevalence of osteoporosis is expected to increase by more than 30% by 2030 [14]. Fractures are one of the most common life-threatening complications of osteoporosis. Osteoporotic hip fractures are associated with short- and long-term disability and mortality rates. According to the data from the National Health Insurance Service in Korea, the one-year mortality rates after osteoporotic hip fractures were 21% in men and 14% in women in 2015, whereas, for vertebral fractures, the one-year mortality rates were 9% in men and 4% in women [15]. If the current aging trend continues, fractures and the associated costs are expected to increase by more than 48%, incurring costs of approximately USD 25 billion by 2025 [16]. Previous studies have reported that women with hyperprolactinemia have a lower spine bone density than healthy controls [17,18,19]. These findings suggest that, beyond infertility, hyperprolactinemia can have multiple effects on bone metabolism, and understanding their relationship is expected to be very helpful in the management of patients with hyperprolactinemia. This review describes the relationship between prolactin and bone metabolism.

Lactotrophs secrete prolactin in a pulsatile manner in the anterior pituitary gland. Prolactin is a polypeptide consisting of 198 amino acids encoded by the *PRL* gene. It exists in various forms depending on its state of aggregation, including little prolactin (22–23 kDa monomer), big prolactin (48–56 kDa homodimer), and big big prolactin (also known as macroprolactin; 100–150 kDa). Among these, the monomeric form is known to exhibit the highest biologic activity [20]. Having a pulsatile secretory manner, its maximum level appears during non-rapid eye-movement sleep, leading to peak serum levels in the early morning [21].

Dopamine is the primary physiological inhibitor of prolactin secretion [22]. Dopamine is released from tuberoinfundibular dopamine neurons in the dorsomedial arcuate nucleus of the hypothalamus, where it acts on the D2 receptors of lactotroph cells in the anterior pituitary gland, thereby inhibiting prolactin synthesis and secretion. This pathway is known as the tuberoinfundibular dopamine pathway (TIDA) [23]. Endothelin-1 and endothelin-3 are also known to inhibit prolactin secretion [24].

Beyond its mammotrophic and lactogenic functions, prolactin also influences the development of neuroendocrine systems and behavioral adaptation [25]. Animal model studies reported that an increase in prolactin levels was associated with enhanced grooming behavior in male rats [26]. Stress responses coping with the academic burden and increased food intake in humans were also reported to have a significant association with prolactin elevation [27,28]. Moreover, healthy lactating women with low anxiety—based on the Hamilton Anxiety Score, a questionnaire used to assess the level of anxiety—appeared to have high levels of prolactin [29,30]. In addition, prolactin has been reported to play a role in immune system regulation because it enhances the thymus function and stimulates T-lymphocyte proliferation in prolactin-deficient mice [31,32,33].

During the later stages of pregnancy and lactation, the inhibitory responsiveness of tuberoinfundibular dopamine neurons to elevated prolactin decreases, allowing prolactin levels to overcome the negative feedback system and remain physiologically high [34]. Prolactin induces the growth of mammary alveoli, which are components of the mammary gland, and promotes milk production [35]. Prolactin stimulates alpha-lactalbumin, the regulatory protein in the lactose synthetase enzyme system, and also stabilizes and enhances beta-casein mRNA transcription [36]. During lactation, prolactin controls adipogenesis by reducing lipoprotein lipase (LPL) activity through the increased expression of prolactin receptors in adipose tissues, while increasing LPL activity in the mammary glands to facilitate lipid production to produce breast milk during lactation [37].

In the hypothalamus, an increase in serum prolactin levels induces the inhibition of GnRH secretion by Kiss1 neurons [38]. Given that GnRH induces FSH and LH synthesis and release, allowing ovulation and menstruation, an interruption in GnRH secretion leads to the inappropriate secretion of LH, and, consequently, to the absence of a preovulatory surge. Moreover, such hypogonadotropic hypogonadism suppresses the progression of the follicular phase of the menstrual cycle, resulting in anovulation and amenorrhea [39,40]. In addition, non-physiological hyperprolactinemia has been reported to have negative effects on various aspects beyond infertility-related issues, such as bone metabolism (Table 2) [41].

## 2. Epidemiology of Osteoporosis and Fractures Due to Hyperprolactinemia

Few studies have investigated the prevalence of osteoporosis and fractures in patients with hyperprolactinemia. In these patients, bone formation markers, such as osteocalcin, decrease, whereas bone resorption markers, including N-terminal telopeptide (NTX), increase, depending on the duration of the disease and prolactin levels [42,43]. This indicates hyperprolactinemia plays a role in stimulating bone resorption, while suppressing bone formation. Schlechte et al. measured forearm and vertebral bone mineral levels in normal women and amenorrheic women with treated and untreated hyperprolactinemia to see if women with PRL-secreting pituitary tumors had similar decreases in cortical and trabecular bones and if bone loss associated with hyperprolactinemia was reversible [17]. The results showed that the spinal bone mineral content in women with hyperprolactinemia was 25% lower than that in healthy women. Women who underwent successful transsphenoidal pituitary surgery for prolactinoma and had regular menstrual cycles had a slightly higher spinal bone mineral content than women with amenorrhea, but this was still lower than that in healthy women. Notably, the forearm bone mineral content was similar to that of healthy women when treated successfully. This indicated that hyperprolactinemia and/or gonadal dysfunction had a more pronounced impact on the trabecular bone in the spine than on the cortical bone in the forearm, and even with successful treatment, it may not have been sufficient to fully restore the bone mineral content to normal levels [17]. In addition, it has been observed that, in fertile female patients with hyperprolactinemia due to pituitary adenomas, the cortical bone density decreases by 17% [44] and the trabecular bone density decreases by 15–30% [17,19,45]. In a case–control study, women with prolactinomas had a higher rate of vertebral fractures than women in the control group [46]. There was no correlation between vertebral and radial bone minerals in women with hyperprolactinemia, and the normalization of estradiol and PRL secretion after successful treatment was not sufficient to restore the bone mineral content to normal. In women with PRL-secreting adenomas, hyperprolactinemia is linked to a high frequency of radiographic vertebral fractures.

According to Biller et al., untreated hyperprolactinemia and persistent amenorrhea are associated with a decrease in bone density due to an estrogen deficiency [18]. In order to compare the bone density between women with treated hyperprolactinemia and normal control women, they divided the participating subjects into five groups as follows: (a) amenorrhea (Group 1): those with no menstrual periods; (b) menstrual recovery after hyperprolactinemia treatment (Group 2): individuals who regained their menstrual cycle after treating hyperprolactinemia; (c) regular menstrual cycles despite hyperprolactinemia (Group 3): those who had regular menstrual cycles despite having hyperprolactinemia; (d) those who had amenorrhea previously but recovered their menstrual cycles with hyperprolactinemia treatment before the study began (Group 4); and (e) oligomenorrhea (Group 5): individuals with infrequent menstrual periods. Groups 1, 2, and 4 exhibited significant initial decreases in lumbar bone density (BD). Group 5 had an initial mean BD that fell between that of amenorrheic and menstruating women, while Group 1 showed a significant decrease in mean BD over 1.7 years. Group 2 showed an increase in BD, although it was not statistically significant. Untreated hyperprolactinemia and persistent amenorrhea are related to an estrogen deficiency, which causes a decrease in bone density.

## 3. Effect of Hyperprolactinemia on the GnRH-LH/FSH Axis

The association between hyperprolactinemia and osteoporosis in women appears to be mediated by an estrogen deficiency due to the suppression of the GnRH-LH/FSH axis in such patients [47]. Klibanski et al. investigated the BMD levels of the spine and hip in 25 women with hyperprolactinemia (13 with amenorrhea and 12 with regular menstrual periods) and 11 with hypothalamic amenorrhea. The mean spinal BMD of women with eumenorrhea and hyperprolactinemia was considerably higher than that of women with hyperprolactinemia and amenorrhea and comparable to that of healthy women [48]. The mean spinal bone density was significantly lower in women with hypothalamic amenorrhea than in those with normal or hyperprolactinemic eumenorrhea. The mean serum estradiol level in women with hyperprolactinemic amenorrhea was significantly lower than that in women with hyperprolactinemic eumenorrhea and comparable to that in women with hypothalamic amenorrhea. These findings demonstrate the correlation between reduced bone density and estrogen deficiency in women with amenorrhea caused by hyperprolactinemia. Bone undergoes continuous remodeling, where bone is resorbed and removed by osteoclasts, followed by the continuous formation of new bone by osteoblasts. This process is referred to as ‘coupling’ [49]. Osteoblasts communicate with osteoclasts through various pathways, and one of these is the RANKL/RANK signaling pathway, which influences osteoclast formation, differentiation, or cell death [50]. Estrogen deficiency is involved in increased IL-7 production, which leads to T-cell activation and an increased production of IFN-γ [51]. These changes in cytokine levels induce the production of the receptor activator of nuclear factor kappa B ligand (RANKL) and TNF-alpha, which promote osteoclast differentiation and bone resorption, ultimately leading to a decrease in bone density [52]. Moreover, an estrogen deficiency causes an iron overload and lipid peroxidation by inhibiting the Nrf2/GPX4 pathway. Subsequently, iron overload and lipid peroxidation cause excessive free Fe^2+^, catalyzing the formation of large amounts of reactive oxygen species (ROS) through the Fenton reaction; an excessive accumulation of endoplasmic reticulum (ER) stress and ROS induce ferroptosis, an important etiology for osteoporosis [53,54].

Changes in BMD were also observed in men with prolactinomas, and this decrease was more prominent in the lumbar spine than in the hip [43,55]. Improvements in bone density in men with both hyperprolactinemia and osteopenia (a T-score between −2.5 and −1.0) was associated not only with the normalization of prolactin levels, but also with the simultaneous normalization of testosterone levels [56]. GnRH suppression due to hyperprolactinemia in men leads to decreased testosterone production by Leydig cells. Similar to estrogen deficiency in women, a testosterone deficiency promotes RANKL production by osteoblasts, leading to osteoclast differentiation [57,58]. Testosterone deficiency also increases IL-6 production, which stimulates osteoclast differentiation and activity, ultimately causing osteoporosis [59].

## 4. Effects of Prolactin on Osteoblasts and Osteoclasts

According to a meta-analysis by D’Sylva et al., both men and women with prolactinomas, regardless of their gonadal function, had significantly lower vertebral fracture rates if treated for hyperprolactinemia than those who were not treated [60]. Among women, untreated patients had vertebral fractures in 46% of the cases, while patients taking cabergoline had fractures in 20% of the cases (OR: 0.29, 95% CI: 0.10–0.78). In untreated male patients, the data showed a 67% rate, compared to a 26% rate in patients treated with cabergoline (OR: 0.18, 95% CI: 0.03–0.94). There was no significant difference observed between gonadal and hypogonadal men (*p* = 0.8). This implied that prolactin could also exert direct effects on bone metabolism in addition to hypogonadism. Patients with untreated hyperprolactinemia had a higher occurrence of fractures compared to those receiving treatment, regardless of their gonadal function. Coss et al. demonstrated that recombinant prolactin and a molecular mimic of phosphorylated prolactin (PP-PRL) were administered to pregnant animals because unmodified and phosphorylated prolactin are the most common types of PRL in rats [47]. The blood samples collected from the female parents exhibited typical levels of estrogen and progesterone, and the additional presence of prolactin did not have any impact on parathyroid hormone, calcium, or alkaline phosphatase. However, newborn pups exhibited a 30% reduction in blood alkaline phosphatase in both groups treated with prolactin, but PTH and calcium levels were comparable to those of the control group. The exposure of primary rat osteoblasts to both prolactins resulted in a decrease in alkaline phosphatase activity, with PP-PRL being the more effective version of the hormone. A histological analysis of pup bone development revealed diminished calvarial bone and decreased endochondral ossification in pups exposed to PP-PRL. This study is the first to demonstrate the direct inhibitory impact of prolactin on the functioning of osteoblasts. In addition, the expression of prolactin receptors was confirmed in osteoblast-like cells, such as MG-63 and Saos cells [61]. Prolactin was also shown to reduce the expression of osteoprotegerin (OPG) proteins in osteoblasts, while increasing the levels of RANKL, a key factor in osteoclast formation. This led to increased bone turnover rates [62] (Figure 1). Seriwatanachai et al. used human pre-osteoblast cells (SV-HFOs) to demonstrate that prolactin significantly reduced osteoblast proliferation while increasing bone formation markers, such as RUNX2 and ALP, during the early stages of osteoblast differentiation. In this study, the effect of prolactin on osteoblasts was independent of the additional supply of sex hormones, providing additional evidence for the direct impact of prolactin on osteoblasts (Figure 2). Notably, in later stages, prolactin reduced the expression of these bone formation markers, indicating a bidirectional effect on osteoblast function [63]. Based on these findings, it is reasonable to believe that prolactin exerts a direct effect on bone metabolism.

Despite the lack of clear evidence of the direct effect of prolactin on osteoclasts, several reports have raised this possibility [64,65]. In addition to RANKL overexpression, prolactin also appears to be involved in the upregulation of osteoclastogenic modulators, such as MCP-1, Cox-2, TNF-α, IL-1, and ephrin-B1, contributing to the regulation of osteoclast function and bone remodeling, which strongly suggests the possibility of its direct effect on osteoclasts [64]. A study using goldfish showed the bidirectional effect of prolactin; it was observed that, at low prolactin concentrations, ranging from 0.01 to 100 ng/mL, osteoclastic activity was reduced, but at concentrations exceeding 100 ng/mL, such activity was increased [65]. However, additional studies on the direct effect of prolactin on osteoclasts are required.

## 5. Protective Effects of Prolactin on Bone

Prolactin has also been reported to exhibit bone-protective effects by directly signaling chondrocytes and synovial fibroblasts to inhibit cartilage degradation, synovial inflammation, and osteoclastogenesis in patients with arthritis [66]. Chondrocytes in articular cartilage express prolactin receptors, and prolactin inhibits the inflammation-induced apoptosis of cultured chondrocytes by preventing p53 induction and reducing the BAX/BCL-2 ratio in response to proinflammatory cytokines [67]. Moreover, prolactin induces the phosphorylation and activation of the signal transducer and activator of transcription-3 (STAT3), resulting in the downregulation of the osteoclast differentiation induced by proinflammatory cytokines, like IL-1β and IL-6, and rheumatoid arthritis (RA) [66]. This includes the suppression of the RANKL receptor activator of the nuclear factor-kappa B ligand, a major promoter of osteoclast formation [66]. As previously mentioned, prolactin was shown to increase bone formation markers, such as RUNX2 and ALP, in the early stages of osteoblast differentiation [63], and osteoclast activity appeared to be reduced at low prolactin concentrations [65]. These findings suggest the protective effects of prolactin on bone metabolism, which seem to be dependent on its concentration or cell stage. Therefore, further research is necessary to better understand the effects of prolactin on bones, which can vary according to external factors.

## 6. Treatment

The primary goal of osteoporosis treatment is the prevention of fractures. Medical therapy is selected for individuals with a history of fractures, those diagnosed with osteoporosis, and individuals with osteopenia who are at an increased risk of fractures. Medications for osteoporosis can be broadly categorized into antiresorptive and anabolic agents. Antiresorptive agents include selective estrogen receptor modulators (SERMs), bisphosphonates (BPs), and RANKL monoclonal antibodies, such as denosumab. SERMs are non-steroidal agents that bind to estrogen receptors, exhibiting both estrogen agonist properties in the bone and brain, and antagonist properties in the breast and endometrium [68]. Raloxifene and bazedoxifene are prescription options available for the treatment of osteoporosis in postmenopausal women. The MORE trial, which included 7705 postmenopausal women with osteoporosis, found that raloxifene reduced the risk of vertebral fractures and increased bone mineral density after 36 months of treatment [69]. Bazedoxifene also has a positive impact on bone mineral density, bone turnover markers, and lipid profile results. Furthermore, it reduces the risk of new vertebral fractures [70]. Bisphosphonates, including alendronate, risedronate, zoledronate, and ibandronate, are available for use. Bisphosphonates should be taken on an empty stomach with water, and patients should remain upright for 30–60 min after taking the medication [71,72]. Alendronate can be administered either once daily or once weekly. In a 10-year randomized controlled trial, treatment with a daily dose of 10 mg of alendronate resulted in an average increase in lumbar spine bone mineral density of 13.7%, 10.3% in the trochanter, 5.4% in the femoral neck, and 6.7% in the total proximal femur over the 10-year period [73]. Risedronate can be administered in various regimens, including 5 mg once daily, 35 mg once weekly, or 150 mg once monthly. In a study involving 2458 postmenopausal women, the group treated with 5 mg/day of risedronate demonstrated a 41% reduction in new vertebral fractures over a period of 3 years compared to the placebo group, and there was an increase in the bone mineral density [74]. Zoledronate can be administered once a year at a dose of 5 mg. In a study on 3889 postmenopausal women, zoledronate was found to lower the risk of vertebral fractures by 70% and the risk of non-vertebral fractures by 25% compared to the placebo group [75]. Ibandronate can be administered orally at a dose of 150 mg once a month or intravenously at a dose of 3 mg every three months. The use of ibandronate in the MOBILE study, which involved 1609 postmenopausal women, resulted in improvements in lumbar spine bone mineral density and proximal femur bone mineral density [76]. Denosumab is administered at a dose of 60 mg once every 6 months [77]. Through the FREEDOM and FREEDOM extension studies, it was confirmed that denosumab administration resulted in increased bone mineral density and a reduced risk of fragility fractures [78,79]. Anabolic agents raise BMD levels by increasing bone formation, and they include teriparatide and romosozumab. Teriparatide is a man-made version of the parathyroid hormone (PTH), which helps bones grow. Romosozumab, on the other hand, is a monoclonal antibody that helps bones grow while reducing bone loss [80,81]. In cases of osteoporosis coexisting with hyperprolactinemia, treatment can involve the use of antiresorptive agents, such as BP or denosumab, along with anabolic agents, such as teriparatide [82,83]. Spironolactone, a competitive antagonist of aldosterone, can reduce oxidative stress in osteoblasts, and can be another possible option for treating osteoporosis in patients with hyperprolactinemia [84]. However, the data on the treatment or prevention of osteoporosis in patients with hyperprolactinemia are still lacking.

## 7. Summary

In conclusion, the decline in the gonadotropin-releasing hormone (GnRH) resulting from heightened levels of prolactin can contribute to a reduction in bone mineral density, a key aspect of osteoporosis. Osteoporosis, characterized by diminished bone mineral density (BMD) and quality, leads to compromised bone strength. In the context of hyperprolactinemia, changes in BMD levels can be triggered either indirectly, through the suppression of the GnRH–gonadal axis caused by elevated prolactin levels, or directly, via prolactin’s influence on osteoblasts and potentially osteoclast cells. The review presented a succinct exploration of the interplay between prolactin and bone metabolism, offering suggestions for further investigation. Additionally, the recent research on bone remodeling is emphasized, shedding light on our evolving understanding of how prolactin modulates these intricate interactions.

## Figures and Tables

**Figure 1 ijms-25-01474-f001:**
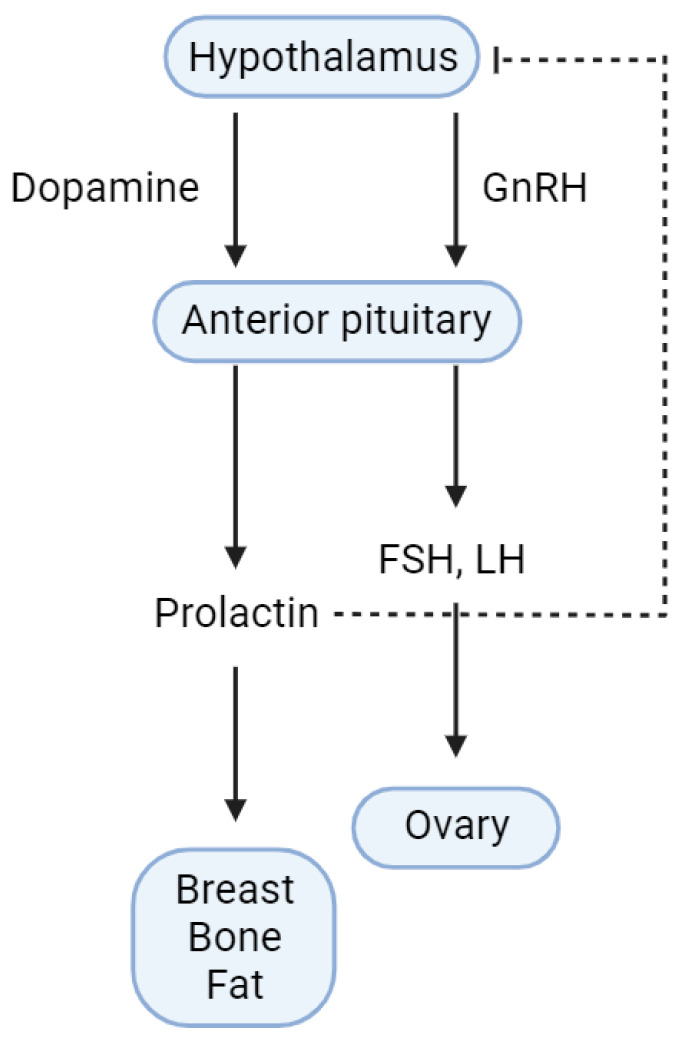
Schematic representation of prolactin signaling pathway in the hypothalamus and anterior pituitary. GnRH, gonadotropin-releasing hormone; FSH, follicle-stimulating hormone; LH, luteinizing hormone. The figure was created with BioRender.com.

**Figure 2 ijms-25-01474-f002:**
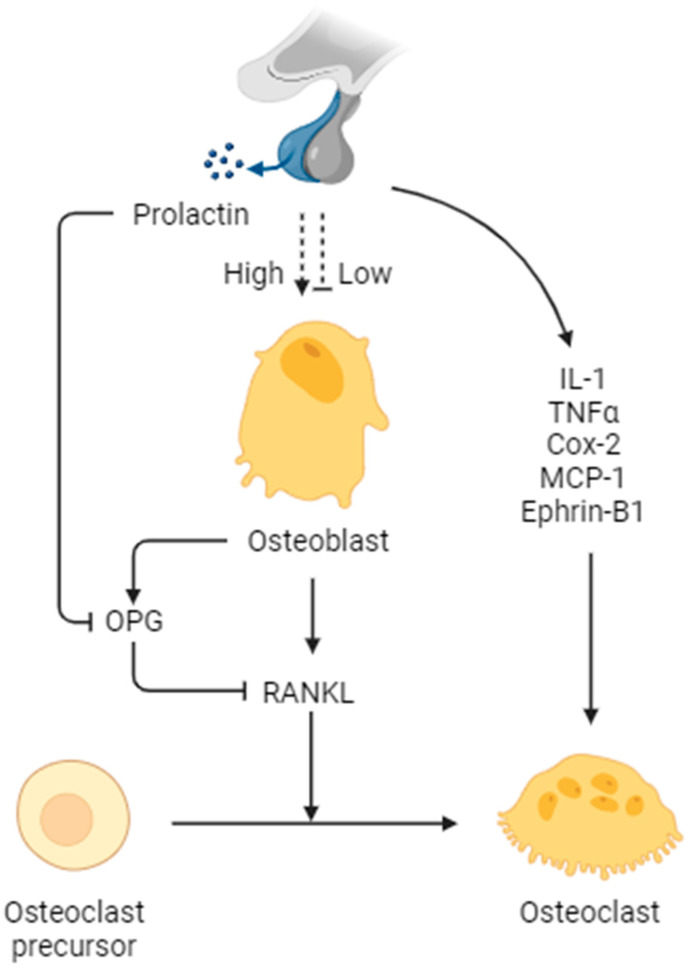
Schematic representation of prolactin signaling in various bone cells. RANKL, receptor activator of nuclear factor-kappa B ligand; OPG, osteoprotegerin; IL-1, interleukin-1; TNFα, tumor necrosis factor α; Cox-2, cyclooxygenase-2; MCP-1, monocyte chemoattractant protein-1. The figure was created with BioRender.com.

**Table 1 ijms-25-01474-t001:** Etiology of hyperprolactinemia.

Disease Origin		
Physiologic	Pathologic	Pharmacologic
Pregnancy	Pituitary tumor	Antipsychotics
Breastfeeding	Hypothyroidism	Antidepressants
Stress	Chronic kidney disease	Antiemetics
Nipple stimulation	Hypophysitis	Antihistamine (H_2_)
Exercise	Polycystic ovarian syndrome	Cholinergic agonists
	Chest wall injury	Domperidone
		Methyldopa
		Metoclopramide
		Verapamil

**Table 2 ijms-25-01474-t002:** Clinical manifestations of hyperprolactinemia.

Signs and Symptoms
Amenorrhea
Low bone mass
Vaginal dryness
Hypogonadotropic hypogonadism
Galactorrhea
Infertility
Erectile dysfunction
Gynecomastia

## Data Availability

Not applicable.

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
