# Peer review of "Effect of Hyperprolactinemia on Bone Metabolism: Focusing on Osteopenia/Osteoporosis"

_ijms, 2024, doi:10.3390/ijms25031474_

Round 1

Reviewer 1 Report

Comments and Suggestions for Authors

The content of this manuscript is interesting. However, there are some improvements that can be made in the manuscript.

Introduction is too superficial and short. It will be good if authors can combine section no 2 and 3 in this section. Also, it will be good if information regarding bone turn over/remodeling i.e some introduction on the bone cells and their function can be included in introduction. As the title mention bone metabolism, hence it is important to include information about this key topic.

Line 22 should be in prolactin paragraph.

Will be good if author can include a diagram showing the overview/link between each hormone and their effects mentioned in introduction (line 40-48)

I would like to suggest authors to reconstruct the flow of this manuscript to have better understanding and flow. For example, can have.

1. intro (with section 2 and 3 combine into this section)

2. epidemiology of osteoporosis and fracture (or any other bone diseases) due to hyperprolactinemia. In this section, can list all studies in a table according to region/continent/counties in the world where studies were conducted too.

3. Effect of hyperprolactinemia on the GnRH-LH/FSH axis

4. Effect of prolactin on osteoblast and Osteoclast

5. Protective effects of prolactin on bone

6. Treatment (author should specify the current treatment for bone diseases induced by hyperprolactinemia) rather than the general treatment for bone diseases. We want to relate to the aim and objective of this manuscript.

7. Summary/conclusion and future direction (this section is missing. strongly suggested that authors should include this section to discuss potential future studies or opportunities).

Comments on the Quality of English Language

English is OK, readable and concise. 

Author Response

Dear Editor and Reviewers,

First of all, thank you for your consideration of publishing our article in the International Journal of Molecular Science.

I enclose our manuscript, which was fully revised based on your valuable requests and comments. The revised parts of our manuscript are colored red. Also, the answers to the reviewers’ comments are included question by question. I hope that our manuscript may become more qualified than the
previous version.

Once again, on behalf of all authors, I truly appreciate your comprehensive comments and look forward to consideration for publication of our revised manuscript.

Sincerely yours,
Sang Ouk Chin
Corresponding author of manuscript No. ijms-2808471

Title: Effect of hyperprolactinemia on bone metabolism: focusing on osteopenia/osteoporosis

Comments from the reviewer 1

The content of this manuscript is interesting. However, there are some improvements that can be made in the manuscript.
Introduction is too superficial and short. It will be good if authors can combine section no 2 and 3 in this section. Also, it will be good if information regarding bone turn over/remodeling i.e some introduction on the bone cells and their function can be included in introduction. As the title mention
bone metabolism, hence it is important to include information about this key topic.

(Answer) As kindly suggested by the reviewer, the authors combined sections 2 and 3 into the introduction.

Line 22 should be in prolactin paragraph.

(Answer) As the reviewer suggested, we moved line 22 to the prolactin paragraph (line 67).

Lactotrophs secrete prolactin in a pulsatile manner in the anterior pituitary gland. Prolactin is a polypeptide consisting of 198 amino acids encoded by the PRL gene. It exists in various forms depending on its state of aggregation, including little prolactin (22–23 kDa monomer), big prolactin (48–56 kDa homodimer), and big big prolactin (also known as macroprolactin; 100–150 kDa).

Will be good if author can include a diagram showing the overview/link between each hormone and their effects mentioned in introduction (line 40-48)

(Answer) The authors added a figure to reflect the reviewer's opinion

Reviewer 2 Report

Comments and Suggestions for Authors

Overall, this is a thorough review of the role of prolactin on bone metabolism.

It also emphasizes the hypothalamic and pituitary pathways in which through prolactin bone metabolism is changed. 

The last section on the treatment of osteoporosis does not seem to fit into the review.  It might be better if cases were discussed in which treatment of the bone to protect from the elevated prolactin levels was discussed.  Otherwise this part of the review does not seem to fit

Author Response

Dear Editor and Reviewers,

 First of all, thank you for your consideration of publishing our article in the International Journal of Molecular Science.

 I enclose our manuscript, which was fully revised based on your valuable requests and comments. The answers to the reviewers’ comments are included question by question. I hope that our manuscript may become more qualified than the previous version.

 Once again, on behalf of all authors, I truly appreciate your comprehensive comments and look forward to consideration for publication of our revised manuscript.

 Sincerely yours,

Sang Ouk Chin

Corresponding author of manuscript No. ijms-2808471

Title: Effect of hyperprolactinemia on bone metabolism: focusing on osteopenia/osteoporosis

 Comments from the reviewer 2

 Overall, this is a thorough review of the role of prolactin on bone metabolism.

It also emphasizes the hypothalamic and pituitary pathways in which through prolactin bone metabolism is changed. 

 The last section on the treatment of osteoporosis does not seem to fit into the review.  It might be better if cases were discussed in which treatment of the bone to protect from the elevated prolactin levels was discussed.  Otherwise this part of the review does not seem to fit

 (Answer) Thank you for your insightful comment on our manuscript. We appreciate your feedback regarding the last section on the treatment of osteoporosis and its perceived lack of alignment with the overall focus of the review. We acknowledge the importance of maintaining coherence in the manuscript.

However, studies on cases or discussions related to the treatment of bone problems associated with elevated prolactin levels are still lacking. Therefore, we introduced bone health treatments in the context of general osteoporosis. 

Your valuable comments have been instrumental in improving the manuscript, and we are committed to providing a more improved and cohesive final version.

Thank you again for your thoughtful review.

Reviewer 3 Report

Comments and Suggestions for Authors

Review note

This paper reviews an important topic – Effect of hyperprolactinemia on bone metabolism.  The authors reviewed a nice set of literatures to address the topic, which will help further translational and clinical application. The structure is generally clear and logically organized. However, to grow into a publication, I think there are some issues the authors need to address.

1.     Osteoporosis is only one common skeletal disorder. The authors only discussed osteoporosis rather than other metabolic disorders in the introduction, paragraph 4, and 5. Is there any correlation between prolactin/hyperprolactinemia and other metabolic bone diseases?

2.     Again, based on the first concern, if the authors would rather be stick with osteoporosis related contents, the authors should make change to the title to be more reflective with the contents.

3.     The authors specifically discussed the direct effect of prolactin on both osteoblasts and osteoclasts, which serve as the mechanisms for hyperprolactinemia-associated osteoporosis. What about the indirect effect? Is inhibition of the GnRH- LH/FSH axis part of it?

4.     I suggest the authors to add-up a Figure(schematics) to help visualize the contents discussed in the treatment section.

In general, this review is clear-cut and reflects an important issue. I hope the author(s) could find some of the above discussions helpful for improving the paper. 

Comments on the Quality of English Language

Minor editing of English language required

Author Response

Dear Editor and Reviewers,

First of all, thank you for your consideration of publishing our article in the International Journal of Molecular Science.

I enclose our manuscript, which was fully revised based on your valuable requests and comments. The revised parts of our manuscript are colored red. Also, the answers to the reviewers’ comments are included question by question. I hope that our manuscript may become more qualified than the previous version.

Once again, on behalf of all authors, I truly appreciate your comprehensive comments and look forward to consideration for publication of our revised manuscript.

Sincerely yours,

Sang Ouk Chin

Corresponding author of manuscript No. ijms-2808471

Title: Effect of hyperprolactinemia on bone metabolism: focusing on osteopenia/osteoporosis

Comments from the reviewer 3

This paper reviews an important topic –  Effect of hyperprolactinemia on bone metabolism.  The authors reviewed a nice set of literatures to address the topic, which will help further translational and clinical application. The structure is generally clear and logically organized. However, to grow into a publication, I think there are some issues the authors need to address.

1.     Osteoporosis is only one common skeletal disorder. The authors only discussed osteoporosis rather than other metabolic disorders in the introduction, paragraph 4, and 5. Is there any correlation between prolactin/hyperprolactinemia and other metabolic bone diseases?

 (Answer) The authors have not been able to elucidate in detail the correlation between hyperprolactinemia and other metabolic bone diseases through the literature and studies presented. There is little research on the potential correlation between hyperprolactinemia and a broader range of metabolic bone diseases. We believe that studying more about the link between high prolactin levels and diseases like Paget disease or certain mineralization disorders will help us get a better idea of how prolactin affects bone health in a range of metabolic disorders.

2.     Again, based on the first concern, if the authors would rather be stick with osteoporosis related contents, the authors should make change to the title to be more reflective with the contents.

 (Answer) As the reviewer suggested, the authors modified the title to better align with the content presented. 

 Effect of hyperprolactinemia on bone metabolism: focusing on osteopenia/osteoporosis

3.  The authors specifically discussed the direct effect of prolactin on both osteoblasts and osteoclasts, which serve as the mechanisms for hyperprolactinemia-associated osteoporosis. What about the indirect effect? Is inhibition of the GnRH- LH/FSH axis part of it?

(Answer) Your comment is right. The authors thought that in the context of hyperprolactinemia related osteoporosis, the indirect effect of prolactin on osteoblasts and osteoclasts was GnRH-LH/FSH axis suppression.

4.  I suggest the authors to add-up a Figure(schematics) to help visualize the contents discussed in the treatment section.

 (Answer) The authors tried to reflect the reviewer’s suggestion, however, we could not add figure due to copyright issues.

In general, this review is clear-cut and reflects an important issue. I hope the author(s) could find some of the above discussions helpful for improving the paper. 

Round 2

Reviewer 1 Report

Comments and Suggestions for Authors

Thank you for the corrections made. The manuscript looks better. However, if authors can suggest better title.

Figure 1 should looks more interesting (i.e organs where the hormones secreted and to see positive feed back and negative feedback)

Author Response

Dear Editor and Reviewers,

First of all, thank you for your consideration of publishing our article in the International Journal of Molecular Science.

I enclose our manuscript, which was fully revised based on your valuable requests and comments. The revised parts of our manuscript are colored red. Also, the answers to the reviewers’ comments are included question by question. I hope that our manuscript may become more qualified than the previous version.

Once again, on behalf of all authors, I truly appreciate your comprehensive comments and look forward to consideration for publication of our revised manuscript.

Sincerely yours,

Sang Ouk Chin

Corresponding author of manuscript No. ijms-2808471

Title: Effect of hyperprolactinemia on bone metabolism: focusing on osteopenia/osteoporosis

Comments from the reviewer 1

Thank you for the corrections made. The manuscript looks better. However, if authors can suggest better title.

Figure 1 should looks more interesting (i.e organs where the hormones secreted and to see positive feed back and negative feedback)

(Answer) The authors modified the figure to reflect the reviewer's opinion.
